# The Underexplored Mechanisms of Wheat Resistance to Leaf Rust

**DOI:** 10.3390/plants12233996

**Published:** 2023-11-28

**Authors:** Johannes Mapuranga, Jiaying Chang, Jiaojie Zhao, Maili Liang, Ruolin Li, Yanhui Wu, Na Zhang, Lirong Zhang, Wenxiang Yang

**Affiliations:** College of Plant Protection, Technological Innovation Center for Biological Control of Plant Diseases and Insect Pests of Hebei Province, Hebei Agricultural University, Baoding 071001, China; jmapuranga@hotmail.com (J.M.); changjiaying@163.com (J.C.); zhaojiaoj918@163.com (J.Z.); maililiang26@163.com (M.L.); m15369870831@163.com (R.L.); wu203474540@163.com (Y.W.); zn0318@126.com (N.Z.); zlr139@126.com (L.Z.)

**Keywords:** wheat, *Puccinia triticina*, leaf rust, resistance mechanisms

## Abstract

Wheat leaf rust, caused by the obligate biotrophic fungus *Puccinia triticina* Eriks. (*Pt*), is one of the most common wheat foliar diseases that continuously threatens global wheat production. Currently, the approaches used to mitigate pathogen infestation include the application of fungicides and the deployment of resistance genes or cultivars. However, the continuous deployment of selected resistant varieties causes host selection pressures that drive *Pt* evolution and promote the incessant emergence of new virulent races, resulting in the demise of wheat-resistant cultivars after several years of planting. Intriguingly, diploid wheat accessions were found to confer haustorium formation-based resistance to leaf rust, which involves prehaustorial and posthaustorial resistance mechanisms. The prehaustorial resistance in the interaction between einkorn and wheat leaf rust is not influenced by specific races of the pathogen. The induced defense mechanism, known as systemic acquired resistance, also confers durable resistance against a wide array of pathogens. This review summarizes the host range, pathogenic profile, and evolutionary basis of *Pt*; the molecular basis underlying wheat–*Pt* interactions; the cloning and characterization of wheat leaf rust resistance genes; prehaustorial and posthaustorial resistance; systemic acquired resistance; and the role of reactive oxygen species. The interplay between climatic factors, genetic features, planting dates, and disease dynamics in imparting resistance is also discussed.

## 1. Introduction

Plants are continuously exposed to a wide range of pathogens and, accordingly, have evolved various defense mechanisms that facilitate early and vigorous pathogen detection and the mobilization of structural and biochemical defenses [1]. Thus, successful plant pathogen infection is more of a rarity than the rule [2]. Biotrophic plant fungal pathogens are the primary cause of most damaging plant diseases, primarily in cereal plants, which results in significant yield losses. Wheat leaf rust, caused by the basidiomycetous fungus *Puccinia triticina* Eriks. (*Pt*), is a macrocyclic foliar disease that continuously threatens wheat (*Triticum aestivum* L.) production in most wheat-growing regions, and can cause significant yield losses over large geographical areas [3,4,5,6,7]. Leaf rust is among the most difficult wheat diseases to control because the pathogen has a high population diversity, the steady development of novel virulence profiles, and the pathogen can strongly adapt to new climate zones [8,9]. The combination of recent adaptations to warmer conditions and the constant rise in global temperature has led to an increase in the occurrence and severity of leaf rust epidemics. Given the vast size of the leaf rust population, it would be reasonable to predict that random mutations would occur with sufficient frequency, resulting in the emergence of new virulent races.

Plants have a fundamental immune system that is constitutively expressed at minimal levels and may be triggered by pathogen infection [10,11]. The manifestation of this immune system bestows inherent degrees of immunity and results in complete or partial resistance against pathogens [11]. Multilayer perceptrons are plants which can incorporate intricate external stimuli and signals throughout their lifespan [12], and their basal immunity can be modulated by both abiotic and microbiotic factors [13,14]. Numerous inducible defense mechanisms have evolved in plants to ward off pathogen attacks. The hypersensitive response (HR), a localized resistance reaction frequently elicited due to pathogen recognition, is distinguished by the expeditious demise of cells at the site of infection [15]. 

The development of resistance to a specific disease is also a result of genetic regulation, race specificity, and longevity. Resistance durability is correlated to the activation of resistance mechanisms based on haustorium formation [16]. Prehaustorial resistance impedes pathogen haustoria formation, even when haustorium mother cells have been normally formed and a papilla is commonly induced at the site where the pathogen is attempting to penetrate the cell wall [17,18]. Prehaustorial resistance is durable, and it does not inhibit the formation of hypersensitive reactions or the sporulating pustules, hence exerting less selection pressure on the pathogen. Posthaustorial resistance arrests the development of the pathogen-feeding structures and terminates the growth of the biomass of the pathogen on the leaf surface. It is also associated with programmed cell death, which is activated during the defense response, as indicated by the accumulation of hydrogen peroxide (H_2_O_2_) at the penetrated epidermal cells [19].

Posthaustorial resistance, which follows the gene-for-gene hypothesis, primarily manifests after haustoria formation, while non-host posthaustorial resistance is activated upon pathogen penetration and remains effective during haustoria formation. Furthermore, the occurrence of non-host posthaustorial resistance is often associated with HR, a regularly observed characteristic that is linked to rare occurrences of non-host penetrations, especially when haustoria initiation takes place [20]. Diploid wild wheats exhibit a spectrum of rust resistance mechanisms, including prehaustorial resistance without necrosis and posthaustorial resistance accompanied by frequent necrosis. Elevated levels of prehaustorial resistance in two out of three *Triticum monococcum* accessions were reported by Niks and Dekens, as well as posthaustorial resistance in *Triticum boeoticum* accession [18]. If the prehaustorial resistance is not sufficiently strong or rapidly expressed, the pathogen will successfully penetrate the host plant cell and establish infection structures, such as substomatal vesicles within the host stomatal cavity. It can then enter the intercellular spaces of the host cells and form infection hyphae, haustorial mother cells, and, sometimes, haustoria, and this may induce posthaustorial resistance [21]. 

Systemic acquired resistance (SAR) is conventionally defined as a plant defense mechanism that confers broad-spectrum, long-lasting resistance to pathogens and has no specificity to the initial localized infection [22]. In many plant species, SAR can be activated by pathogens that cause necrosis, either as a component of HR or as a disease symptom. It is distinguished by elevated concentrations of the hormone salicylic acid (SA) within the tissues of plants [23]. Both the range of pathogen protection and the concomitant changes in gene expression distinguish SAR from other disease responses. Reactive oxygen species (ROS) have direct antibacterial properties and also serve as crucial signals in the activation of plant innate immune responses. Strict regulation of ROS levels is necessary to prevent cellular damage resulting from its excessive accumulation. The production of ROS during PTI is mediated by members of the nicotinamide adenine dinucleotide phosphate (NADPH) oxidase family [24]. In plants, NADPH oxidases are often referred to as respiratory oxidase homologs (RBOHs) and are responsible for the production of O^2−^ in extracellular spaces. ROS produced by RBOHs serve as secondary messengers for the fast transmission of both local and long-distance signaling [25]. This article provides an insight into the mechanisms of wheat resistance to leaf rust.

## 2. Host Range, Pathogenic Profile, and Evolutionary Basis

### 2.1. Host Range and Pathogenic Profile

*Pt* exhibits heteroecious characteristics, necessitating the presence of both a telial/uredinial host, often wheat, and an alternate host, such as *Thalictrum speciosissimum* or *Isopyrum fumaroides*, in order to successfully complete its whole life cycle [3]. Its primary hosts include hexaploid common wheat (*T. aestivum*), wild emmer (*T. dicoccoides*) [26], tetraploid durum (*T. turidum* spp. *durum*), domesticated emmer wheat (*Triticum dicoccon*), triticale (X *Tritico secale*), *Aegilops speltoides*, and common goatgrass (*Ae. cylindrica)* [26,27]. *Pt* is a macrocyclic and heteroecious rust fungal species that has five distinct spore stages and two host species that are taxonomically unrelated. The sexual cycle occurring on an alternative host also contributes to the generation of genetic variety in the pathogen via genetic recombination [28,29]. The urediniospores produced on wheat hosts exhibit a dikaryotic state, possess a width of 20 µm [30], and they can spread over a long distance with airflow and reinfect the telial host multiple times under favorable conditions, especially when exposed to water on leaf surfaces at temperatures ranging from 10 to 25 °C [3]. 

### 2.2. Evolutionary Basis of Host–Pathogen Interactions

To gain comprehensive insights into the disease state and formulate effective disease management strategies, it is crucial to comprehend the origins, variability, distribution, and routes of movement of the rust pathogens [31,32,33,34]. The evolutionary potential of these pathogens is significantly influenced by the genetic structure within their populations and their capacity for fast intercontinental dissemination. The comprehension of the genetic structure of phytopathogen populations might provide valuable insights for the formulation of an optimal breeding approach aimed at developing durable resistance to leaf rust [35]. It is evident that plant–pathogen interactions exhibit spatial and temporal variability, indicating that they do not progress uniformly [36,37,38]. The interactions between plants and microbes serve as a paradigmatic example of rapid evolution [39], where the molecular conflict between plants and microbial invaders plays a crucial role in the establishment of advantageous allelic variations within the genomic pool [4]. Pathogens derive advantages from modifications that facilitate their evasion or suppression of plant defenses, while plants reciprocally gain from innovations that enhance their immune capabilities [40]. Evolutionary genomic studies serve the purpose of elucidating the origins of pathogen lineages and the spatial distribution of genetic diversity, while also providing insights into the manner in which natural selection shapes genetic variation across the whole genome [41].

The battle between plants and pathogens causes perpetual co-evolutionary cycles [42,43,44]. Allele frequency fluctuations in both host and pathogen populations are driven by negative frequency-dependent selection, hence serving as a mechanism for the maintenance of genetic variety within both populations [45]. The interaction between hosts and pathogens may be characterized as an ongoing arms race. Hosts are subject to selective pressures that favor the elimination of pathogens, while pathogens, in turn, undergo evolutionary changes to elude host immunity. Directional selection is a significant driving force in the evolutionary dynamics between hosts and pathogens. Specifically, hosts experience selective pressure to minimize their interactions with pathogens, whereas pathogens face selection to enhance their interactions with hosts [45]. Different patterns of polymorphisms are likely to arise via frequency-dependent selection and arms-race dynamics [46]. Alleles may persist for a long time, and genetic variation can be detected in natural populations due to stable polymorphism. Polymorphism can give rise to specialized pathogens that are capable of infecting a restricted set of host phenotypes, while polyphenism has the potential to give rise to generalist pathogens that are capable of infecting a broader variety of hosts [47]. Co-evolution, wherein the fitness of genotypes of one species is determined by the gene frequencies of other species, is driven by this confrontation, which also generates diversity in host defenses and pathogen weaponry [48].

## 3. Wheat Infection by *Pt*

### 3.1. Penetration and Colonization of Wheat by Pt

The infection process of leaf rust fungi start with spore germination, the directional growth of the germ tube towards a stoma, the differentiation of an appressorium over the stoma, and penetration into the substomatal cavity. Apparently, features of the plant such as the stomatal guard cell morphology [49] and epidermal cell deviating micromorphology [50] may reduce the infectibility by lowering the amount of infection units (appressoria over stomata). Furthermore, even if an appressorium is eventually established over a stoma to penetrate the intracellular spaces of the mesophyll cells, potentially bypassing epidermal responses to produce the infection peg, germlings that take longer to locate a stomata will be depleted of reserves, hence reducing their probability of successfully forming a haustorium [51]. Haustorium formation occurs in the wheat infected cells, and the haustorium is not only a sophisticated structure for nutrient acquisition, but is also an intense site where proteins, including effectors, are secreted into the host to suppress host immunity. Transcriptomic analyses of all six races of *Pt* isolates identified 456 haustorial secreted proteins [52]. *Pt* whole-genome sequencing found more than 600 significant annotated proteins that possess a secretory peptide [53]. A recent transcriptome analysis of *Lr19*-virulent mutants identified eight secreted proteins that were *AvrLr19* candidates [54]. Although various candidate secreted effector proteins have been identified, only a few *Pt* effectors have been successfully cloned (Table 1), including Pt18906 [55], Pt13024 [56], and Pt_21 [57]. Pt13024 was shown to suppress programmed cell death and trigger the accumulation of ROS and callose deposition [56]. The recently characterized wheat leaf rust fungus effector, Pt_21, was found to suppress host defense responses by directly targeting wheat TaTLP1 and inhibiting its anti-fungal activity [57]. This clearly demonstrates that the secreted effectors enhance pathogenicity by manipulating the functions of the host plant targets or suppressing the host defense responses by either functioning as enzymes or through other roles. 

### 3.2. Molecular Basis Underlying Wheat–Pt Interactions

Plants have evolved intricate and protective surveillance networks which constitute the induced defense response to effectively counteract and safeguard against pathogenic microbes [10]. The first layer of the plant innate immunity is PAMP-triggered immunity (PTI), which is activated via the recognition of pathogen-associated molecular patterns (PAMPs) or microbe-associated molecular patterns (MAMPs) by pattern-recognition receptors (PRRs) [60,61,62]. PTI activation triggers multiple signaling pathways in the host cells, which include a rapid increased influx of extracellular Ca^2+^ into the cytosol, the activation of MAPK pathways, ROS signaling, other signaling molecules like SA and n-hydroxypipecolic acid, the expression of defense responsive genes, stomatal closure, and callose deposition [63,64,65,66,67,68,69]. These responses curtail pathogen invasion and colonization. To function as pathogens, microorganisms must effectively suppress or subvert their host plant’s defense responses. Plant fungal pathogens secrete virulence effector proteins which manipulate host immunity, resulting in the plants being more susceptible to disease, a process known as effector-triggered susceptibility (ETS) [10,70], which is again counteracted by the host through a second layer of defense known as the effector-triggered immunity (ETI) (Figure 1). PTI and ETI entail the activation of two unique receptor classes (i.e., PRRs and nucleotide-binding leucine-rich repeat receptors (NLRs), respectively) and different stages in early signaling [70,71]. Despite the fact that PTI and ETI are from different layers of the plant immune system, they share many genes and pathways of the immune signaling pathway network [72]. 

PRR signaling potentiates ETI, demonstrating that cell priming by PRRs results in the subsequent activation of defense by NLRs [69,73]. Furthermore, a globally similar transcriptional output is activated by both PTI and ETI, with an enhanced PTI being induced by NLRs [74,75,76]. Subsequently, the synergistic action of pathogen effectors that suppress host defense responses theoretically affect both PTI and ETI [77]. To subvert these constraints, instead of directly recognizing pathogen effectors, host plant NLRs constantly monitor the intracellular milieu to detect the suppression of the host defense system by these effectors. This subsequently triggers the activation of NLRs that are associated with the virulence target [70]. This implies that the activation of NLRs necessitates an attempted suppression of PTI [78]. The restoration of an effective immune response by NLRs in the presence of pathogen effectors is still a major enigma in plant pathology. It was previously postulated that ETI signaling might overcome ETS by inducing a more robust PTI response, which results in the renewal of PTI signaling components, hence facilitating the induction of immunity [69,73].

The interaction between wheat and *Pt* is also characterized by a conventional direct gene-for-gene system [79] or receptor–ligand model, making it an excellent model for investigating plant–pathogen interactions. The gene-for-gene hypothesis states that such an interaction is determined by the presence of a single plant-dominant *R* gene, which allows the direct or indirect recognition of pathogens producing specific proteins (codified by avirulence (*Avr*) genes) [80,81]. This interaction results in a compatible or incompatible interaction. A compatible interaction is achieved when a pathogen develops and reproduces without an active host resistance. An incompatible interaction occurs as a result of the development of a resistance allele towards the pathogen, resulting in the failure of the pathogen to effectively proliferate due to the combination of *R* genes and *Avr* genes. Host plant R proteins detect Avr factors of the pathogen and activate signal transduction cascades, causing hasty defense activation. While the *Avr* genes and *R* genes found in *Pt* and wheat, respectively, exhibit specificity in their interactions, it should be noted that these genes do not always display dominance and do not always engage in a one-for-one relationship [3]. The mechanisms of host–pathogen interactions has been intensively reviewed, and different proposed models to interpret the mechanism of how the NLR proteins recognize pathogen effectors have been compared [82,83,84,85,86].

## 4. Mechanisms of Wheat Resistance to Leaf Rust

Wheat resistance to leaf rust can be divided into race-specific resistance and non-race-specific resistance based on genetic determinations, physiological features, and molecular mechanisms. Race-specific resistance is also called qualitative resistance or major gene resistance. It follows the gene-for-gene hypothesis [79] and is characterized by the presence of HR and induction of rapid cell death at the infection sites [3,87]. Non-race-specific resistance is also called adult plant resistance (APR), quantitative resistance, or slow-rusting resistance [88]. Histological observations revealed another two types of resistance mechanisms in wheat, namely prehaustorial and posthaustorial resistance [18,89,90]. Although many review articles have been published on the genetics of wheat resistance to rust [91,92,93,94,95,96,97,98,99], no review paper has been published that specifically focuses on haustoria formation-based resistance. Various studies have reported prehaustorial and posthaustorial resistance mechanisms against leaf rust in hexaploid wheat, but these studies have been mainly based on histopathological observations in which they were focusing on the fungal growth and development, specifically the haustorium [16]. Therefore, there is a need for the elucidation of the genetic background of haustorium formation-based resistance and how it is inherited. A recent study identified the genomic regions and candidate genes associated with prehaustorial resistance in *T. monococcum* [100]. Therefore, this section will firstly discuss the wheat leaf rust resistance (*Lr*) genes that have been cloned and characterized so far and then prehaustorial and posthaustorial resistance, as well as systemic acquired resistance and the role of ROS in wheat resistance against leaf rust.

### 4.1. Cloning and Characterization of Wheat Lr Genes

More than 100 leaf rust resistance (*Lr*) genes (~50% derived from wild progenitor and non-progenitor species) have been discovered, and only a few of these have been cloned so far [97,101]. The feasibility of cloning *Lr* genes or leaf rust resistance QTL (*QLr*) has been enhanced by advances in genomic sequencing and molecular biology techniques. Multiple strategies including classical map-based positional cloning or rapid gene-cloning approaches like MutRenSeq, AgRenSeq, MutChromSeq, and MutIsoSeq, as well as whole-genome sequencing, can be utilized to clone these genes [102,103]. So far, only eleven wheat *Lr* genes have been cloned (Table 2), either via classical map-based cloning, (*Lr1* [104], *Lr10* [105], *Lr21* [106], *Lr34* [107], *Lr42* [108], and *Lr67* [104,105,106,107,108,109]) or through rapid gene-cloning approaches such as MutRenSeq (*Lr13*) [110,111], TACCA (*Lr22a*) [112], MutChromSeq (*Lr14a*) [113], and MutIsoSeq (*Lr9/Lr58*) [114]. Most of the *Lr* genes that have been cloned, including *Lr1*, *Lr10*, *Lr13*, *Lr21*, *Lr22a*, and *Lr42*, are race-specific resistance genes that encode nucleotide-binding site leucine-rich repeat (NLR) proteins [104,105,106,108,111,112]. In addition to the NLR proteins, *Lr14a* is another race-specific resistance gene that encodes a membrane-localized protein with twelve ankyrin repeats and Ca^2+^-permeable non-selective cation channels [113]. *Lr9/Lr58* is also a race-specific resistance gene which encodes a tandem kinase fusion protein [114]. *Lr34* and *Lr67* are known as slow-rusting genes or adult plant resistance genes, encoding a putative ATP- binding cassette (ABC) transporter and a hexose transporter, respectively [107,109].

### 4.2. Prehaustorial Resistance

Induced defense is prompted by the infection attempt of the respective pathogen. In rusts, one particularly common resistance feature is defective haustorium formation on non-host plant species and some resistant host species, termed prehaustorial or pre-cell-wall penetration resistance [116], and it is characterized by the expression of resistance prior to fungal penetration into the host. It was firstly described as reduced fungal penetration into the host during the interactions between rust fungi and non-hosts [17], and later, it was described in detail during the interactions between barley and leaf rust [89]. When partially resistant barley leaves were inoculated with leaf rust fungus *Puccinia hordei*, prehaustorial resistance was described as the inability to form a haustorium on the barley leaves. The inability to form haustorium results in a delayed progression of rust fungus growth (prolonged latency phase) and reduced or no spore production (no sporulation). Functional targets of prehaustorial resistance mechanisms are germ tubes and appressoria (ectophytic phase of pathogenesis), as well as substomatal vesicles and infection hyphae (early endophytic phase of the pathogenesis of rust fungi). This defense response typically leads to the formation of cell wall reinforcements, also called cell wall appositions or papillae [117,118]. Papillae were observed at the site of penetration and were shown not only to be involved in inhibiting pathogen penetration, but have also been implicated in repairing the cell wall subsequent to pathogen penetration attempts [119,120]. 

Mounting evidence has recently reported the participation of some wheat *R* genes in conferring prehaustorial resistance mechanisms that stop the development of rust fungi at the early endophytic and even ectophytic stages. It was reported that *Lr1*, *Lr3a*, *Lr9*, *LrB*, *Lr19*, *Lr21*, and *Lr38* are involved in prehaustorial resistance to leaf rust [121]. The prehaustorial resistance level in *T. monococcum* seedlings was found to be higher than in Thatcher-*Lr34* seedlings [119,122]. Wheat defense responses, including oxidative bursts and micronecrotic reactions associated with pathogen infection, were activated by the interaction between *Pt* and *TcLr9,* resulting in the complete termination of pathogen development [123]. It was found that both resistant *TcLr9* and Thatcher plants support the early stages of pathogen growth, including the germination of urediniospores, appressoria formation, and haustorial mother cell development 124]. The suppression of *Pt* in *TcLr9* starts after the development of haustoria, despite the fact that real recognition occurs quickly after the formation of appressoria; as a result, the resistance is referred to as prehaustorial resistance. Prehaustorial resistance was associated with callose deposition and cellular lignification in the vicinity of the penetration site, and HR induction associated with the death of adjacent infected cells was also reported [124]. The prehaustorial resistance to wheat leaf rust discovered in the diploid wheat einkorn (*T. monoccocum* var. *monococcum*) accession PI272560 confers race-independent resistance against isolates that are virulent on accessions harboring resistance genes located on the A-genome of *Triticum aestivum* [125]. The establishment of prehaustorial resistance in accession PI272560 resulted in the abortion of fungal development during the formation of haustorial mother cells and the production of higher levels of H_2_O_2_ compared to the susceptible accession 36554 (*T. boeoticum* ssp. *thaoudar* var. *reuteri*) [125]. 

Some resistance mechanisms and transcriptome alterations were reported to be occurring in the background of PI272560 prehaustorial resistance because an increase in levels of phenolic substances and chitinase activity at the infection site, as well as pathogenesis-related genes, was observed at 24 h post-inoculation (hpi) compared to *T. boeoticum* accession (36554) [125]. It was recently found that a gene (TuG1812G0500002899) located on chromosome 5A, which encodes berberine bridge enzyme (BBE)-like Cyn d 4, was highly expressed at 8 hpi in PI272560 compared to the partially susceptible 36554 [100]. Serfling and colleagues reported that the BBE may instigate and trigger hypersensitive cell death [125], implying that it might be a critical enzyme for basal defense responses [126], but is also a key enzyme in non-host resistance [127,128]. As a progenitor of wheat, *Triticum urartu* is closely related to einkorn (*T. monococcum*), which often displays a high level of resistance to wheat leaf rust. Therefore, einkorn can be a useful resource for breeding pathogen-resistant wheat varieties in the future. 

*Pt* infection was found to be inhibited by the hypersensitive prehaustorial effector-induced immune reaction in einkorn accession PI272560. Surprisingly, this effector-induced immune response is non-race-specific (horizontal), which renders it atypical. Only a few non-race-specific resistance genes are known, including *Lr34*, *Lr46*, and *Lr67,* which are only active during the adult phases of plant development [3]. The resistance imparted by *Lr34* is distinguished by the absence of chlorosis and necrosis on flag leaves, as well as fewer and smaller uredinia, rather than a hypersensitive response [129]. During prehaustorial resistance, the formation of haustoria is frequently impeded prior to the development of fungal sporelings, which is caused by callose deposition at the site of cell wall penetration [130]. Similar to the quantitative resistance provided by *Lr34*, prehaustorial resistance is believed to confer resistance to *Pt* that is not specific to any particular race. However, it was observed that, in the majority of cases, prehaustorial resistance does not manifest as visible necrosis at the macroscopic level in *T*. *monococcum* accessions [119]. *Lr67* has comparable traits to *Lr34* in providing partial or slow-rusting, non-race-specific or broad-spectrum APR to leaf rust and stem rust. *Lr67/Yr46/Sr55/Pm46/Ltn3* has been shown to provide partial resistance against leaf rust, stem rust (*Sr55*), stripe rust (*Yr46*), and powdery mildew (*Pm46*), and is associated with leaf tip necrosis (*Ltn3*) [131,132]. Since *Lr34* and *Lr67* confer non-race-specific resistance, which is not characterized by necrosis, it implies that these two genes impart prehaustorial resistance. A detailed molecular characterization of APR genes in wheat, as well as an understanding of their functionality and interactions when multiple APR and *R* genes are stacked in a single genotype via conventional and genetic modification breeding, is a research priority that will contribute to the understanding of leaf rust resistance breeding [91]. Mostly, biotrophic fungal pathogens like *Pt* are strictly host-specific. Thus, single genes from alien species could potentially introduce durable leaf rust resistance in wheat cultivars. 

### 4.3. Posthaustorial Resistance

Posthaustorial resistance, also known as post-cell-wall penetration resistance, allows pathogen penetration into the host cells and the formation of haustorium by invaginating the host mesophyll cells [18,119]. Posthaustorial resistance is induced by the formation of at least one haustorium or, sometimes, the successful formation of pathogen colonies [124]. During posthaustorial resistance, the plant cell containing haustoria often dies, hence impeding the invasive growth and proliferation of the pathogen in adjacent cells, and this defense response is known as HR [18,133]. In general, race-specific hypersensitivity resistance is posthaustorial [119]. In host–pathogen interactions in which the host plant harbors race-specific resistance, the induction of HR occurs immediately after haustorium formation inside the host cells [17,119]. HR is also involved in the synthesis of secondary metabolites, the production of pathogenesis-related (PR) proteins, cell wall reinforcements, and in some cases, it is followed by ROS accumulation at the plasma membrane. Consequently, this results in an increased influx of Ca^2+^ levels, transcriptional reprogramming, and the activation of protein kinase cascades. Increased levels of H_2_O_2_ accumulation have been shown to trigger a signaling pathway that subsequently induces robust PCD [134]. During non-host interactions, the manifestation of posthaustorial resistance might occur at the position where haustoria formation occurs, and it may be enclosed in callose, leading to HR on the host cell. One notable distinction between gene-for-gene posthaustorial resistance and non-host posthaustorial resistance lies in their respective timing. Gene-for-gene posthaustorial resistance primarily manifests after the formation of haustoria, while non-host posthaustorial resistance initiates following pathogen penetration and persists throughout haustoria formation [20].

### 4.4. Systemic Acquired Resistance

Systemic acquired resistance (SAR) is a broad-spectrum resistance in plants that involves cellular processes such as the recognition of PAMPs or effectors, the transcriptional activation of battery of *PR* genes, MAPK signaling, and HR. The signal transduction pathway leading to SAR is principally regulated by *NPR1* (*non-expressor of PR genes*), also called *NIM1* (*non-inducible immunity 1*) [135,136,137,138,139,140]. SAR expression primarily depends on SA, a signaling molecule which triggers the expression of defense genes via NPR1 [141]. SA is a critical signaling phytohormone that is essential for activating several host plant immune responses against biotrophic and hemi-biotrophic fungal pathogens, while the response to necrotrophic fungal pathogens involves primary hormones JA and ET [142]. In addition to SA, N-hydroxypipecolic acid (NHP) and its precursor, pipecolic acid, serve a critical role in signaling plant immunity and are needed for SA biosynthesis [143,144]. Pathogen invasion activates SAR, which is expressed as cell death responses varying from single-cell HR to necrotic disease lesions [145]. In the absence of pathogen infection, NPR1 oligomerizes in the cytoplasm through intermolecular disulfide bonds [146]. However, during pathogen infection, redox changes occur in the cytoplasm which trigger the reduction of disulfide bonds, and monomeric NPR1 translocates into the nucleus, where it serves as a transcriptional co-activator at the target gene promoter, thereby activating defense gene expression [146,147,148]. NPR1 regulates the expression of genes by physically interacting with TGA transcription factors via the ankyrin repeats, which bind to *PR* gene promoters to activate expression in the presence of SA and repress expression in the absence of SA [149,150,151,152]. TGA2, TGA5, and TGA6 have been shown to function superfluously in SA-induced *PR* gene expression and disease resistance (Figure 2) [152]. The NPR1 C-terminal domain is essential in SA binding and transcriptional activation [138]. Defense gene expression is prevented by the inhibition of *NPR1* reduction, whilst constitutive monomerization, nuclear localization, and defense gene expression are a result of Cys82 or Cys216 mutation in NPR1 [146]. 

NPR4 and its close homology NPR3 have functional redundancy in negatively regulating plant immunity [153]. Downstream of SA, two parallel signaling pathways have been hypothesized in recent research on NPR1, NPR3, and NPR4 [140]. On one hand, NPR3 and NPR4 suppress defense gene expression when SA levels are low, but when SA levels rise due to pathogen infection, NPR3 and NPR4 activities are reduced, and the transcriptional repression of SA-responsive genes is relaxed [140,154]. Pathogen-induced SA accumulation, on the other hand, enhances the transcriptional activation activity of NPR1 to further induce the expression of defense-related genes (Figure 2). Intricately woven together, they provide fine-tuned modulation of the defense response to varying levels of SA. A comprehensive review of the contribution of SA signaling via NPR1 and NPR3/4 to local and systemic defensive responses was also conducted by Liu and colleagues, and they established that most SA-triggered immune responses in plants require both types of SA receptors [155]. *PR1*, *PR2,* and *PR5* are the *PR* genes that were demonstrated to be induced by SA, and these are also used as SA signaling markers [22]. 

The recognized function of the NPR1 protein in modulating SAR via the expression of *PR* genes has been documented in several plant species, such as *Arabidopsis* and rice, among others, under diverse biotic stress conditions [156]. Nine *NPR1* homologues (*TaNPR1*) were identified in bread wheat [157,158], and it was recently postulated that *NPR1* negatively regulates wheat resistance to stem rust infection by functioning at the *Ta7ANPR1* locus via an NB-ARC-NPR1 fusion protein [158]. However, the role of the wheat *NPR1* homologues in wheat defense responses against leaf rust remains a mystery. A potentially more sophisticated host might include using *TaNPR1* as a decoy in the detection, while also developing an alternate mechanism to circumvent *TaNPR1* in instances when the integrity of the *NPR1* component is disrupted by pathogens [158]. The comprehensive understanding of the functions of wheat *NPR1* homologues in wheat resistance to leaf rust may facilitate the development of approaches to alter the interactions between wheat and *Pt* through the modification of the expression of the respective genes, utilizing transgenic or genome-editing technologies like CRISPR/Cas9 to impart broad-spectrum resistance in wheat [159]. 

## 5. ROS and Their Role in Wheat Defense against Leaf Rust

Several plants respond to pathogenic fungal infection with ROS as a strategy to apprehend fungal growth. ROS production is among the initial cellular reactions that occur after successful pathogen recognition. The apoplastic production of superoxide or H_2_O_2_ has been observed in response to the detection of several pathogens [160,161,162]. ROS are produced as aerobic respiration by-products in chloroplasts, mitochondria, glyoxysomes, and peroxisomes, as well as being generated as a sovereign product, and were originally considered harmful towards the cellular macromolecules [163]. However, ROS molecules were recently demonstrated to be essential for plant defense against various stresses after being involved in indispensable defense mechanisms like HR that result in programmed cell death, as well as SA-mediated signal transduction pathways [164]. The successful recognition of avirulent pathogens by host disease resistance proteins induce a two-phase accumulation of ROS. The first phase is characterized by a low-intensity and transient ROS accumulation, while the second phase is sustained and exhibits a significantly higher magnitude, which is associated with disease resistance [165]. However, pathogenic microorganisms that are highly infectious and manage to evade detection by the host’s immune system only trigger a short-lived and very weak first immune response, implying that ROS may play a crucial role in establishing host defense responses. PAMPs also induce an oxidative burst. Within the host plant cell, ROS can directly induce the reinforcement of host cell walls through glycoprotein cross-linking or lipid peroxidation and the subsequent impairment of the cellular membrane [165,166]. However, it is apparent that ROS are critical in the activation of defense-related genes [167]. ROS play a role in defense mechanisms alongside other plant signaling molecules, such as SA and nitric oxide (NO). However, it has been observed that ROS also serve an important role in the regulation of other plant responses relative to other signals.

ROS serve as antimicrobials, plant cell wall cross-linkers to block pathogen entry, and local systematic secondary messengers to activate further immune responses like stomatal closure or gene expression [25,165,168,169,170,171]. Invaded plant cells are directed towards apoptosis by sufficient concentrations of ROS to curb the spread of biotrophic fungal infection through nutrient supply depletion [163]. During plant innate immunity, the nicotinamide adenine dinucleotide phosphate (NADPH) oxidase RBOHD (respiratory oxidase homolog) is a prime player in the production of ROS [24]. ROS production is one of the key preliminary responses that is activated immediately after PAMP detection. For a successful immune response outcome, there is a need for the precise regulation of the timing, amplitude, and duration of the induced response. This regulation takes place at different levels, including the primary receptor complex, downstream signaling components, and transcriptional regulators. Unprompted instigation or the inability to inhibit signaling after immune activation cause detrimental effects on the host [172]. ROS production also occurs during ETI, but at a lower stride. The production of ROS during PTI and ETI principally depends on RBOHD (Figure 1). An Arabidopsis RBOHD mutant was demonstrated to be deficient in ROS production upon PAMPs detection, but in response to pathogen infection, barley produces ROS and triggers ETI [134,173,174]. The accumulation of oxidative bursts that include O^2−^ and H_2_O_2_ occurs as early signaling molecules during pathogen infection [175]. 

Mellersh et al. (2002) described the significance of plant defense strategies against fungal penetration in plant–fungus interaction. The importance of H_2_O_2_, superoxide, and phenolic compounds in curbing fungal growth and development was highlighted [176]; nevertheless, biotrophic fungi have evolved some strategies to surmount increased intracellular ROS levels [177]. ROS maintenance is under a subtle relationship in host plants through the regulation of its production and protection from its lethal effects [163]. Detoxification or scavenging can be used to regulate ROS [178]. Light microscopy was used to study the role of ROS in the defense of wheat near isogenic lines harboring genes resistant to leaf rust introgressed from a wild species [179]. Superoxide ions were observed in the infection sites. The development of *Pt* in the near-isogenic line harboring the *Lr38* resistance gene induced oxidative bursts at the sites of infection, which curtailed pathogen invasive growth. The inhibition of oxidative bursts in near-isogenic lines harboring *Lr38*, *Lr19,* and *Lr24* resulted in a reduced HR development [179]. The accumulation of ROS in wheat near-isogenic line TcLr27+31 was also reported to be induced by *Pt* effector Pt18906 [55]. It was also recently found that *Pt* effector Pt13024 greatly induced ROS accumulation in *TcLr19* and a higher callose deposition in the wheat near-isogenic line *TcLr30* [56]. 

## 6. Interplay between Climatic Factors, Genetic Characteristics, Planting Dates, and Disease Dynamics

Changes in temperature and other climatic factors, such as modified patterns of rainfall, can lead to a range of changes related to wheat pathogens, which typically include geographical distribution, seasonal phenology, and population dynamics [180]. Alterations in climatic conditions can influence the seasonal timing of pathogen life cycle events, potentially causing them to coincide with specific stages of host plant growth or the presence of natural antagonists or synergists. Furthermore, changes in the climate can impact the population dynamics of wheat pathogens, affecting factors such as over-wintering and survival rates, infection efficiency, and the duration of the latency period. This may ultimately lead to changes in the occurrence and severity of diseases within a certain geographical area [181]. The prediction of potential consequences of climate change on global wheat production is highly intricate and challenging due to a limited comprehension of the interactions between multiple abiotic factors, encompassing temperature, precipitation, and ambient levels of various atmospheric gases, such as carbon dioxide, ozone, and others [182]. Wheat genotypes and climatic factors were reported to have a significant influence on the interactions among obligate biotrophic pathogens and the predominance of one pathogen over another [183]. 

While host genotype resistance and the application of fungicides are commonly used methods for managing wheat rusts, there is a need for effective agronomic strategies to minimize disease management expenses and enhance the sustainability of wheat production. A comprehensive understanding of the spatial and temporal heterogeneity in the structure of wheat rust development is expected to contribute to improved efficacy and sustainability in disease management strategies [184]. For instance, wheat leaf rust outbreaks were exacerbated by the integration of delayed planting and maturation, earlier disease onset, reduced cultivar resistance, elevated winter temperatures, and an increased frequency of cold and wetter days throughout the autumn–winter–spring period [185,186]. The aforementioned significant associations also underscored the possibility for enhancing levels of resistance in wheat cultivars through the strategic selection of appropriate planting dates, hence mitigating disease development under favorable climatic conditions. Nevertheless, the current understanding of the integrated interplay between climatic factors, disease dynamics, genetic characteristics, and planting date in relation to wheat leaf rust intensity prediction remains limited. There were substantial differences in the severity of leaf rust between different cultivars, planting dates, period of sampling, and year parameters [186]. It was suggested that including cultivar resistance, wheat maturity dates, planting practices, and weather predictors into research on wheat leaf rust may enhance the predictive accuracy of future models, improve the longevity of host resistance to the disease, and promote the sustainability of disease management strategies. 

The potential enhancement of host resistance to pathogens might be attributed to the optimization of both static and dynamic defense mechanisms following modifications in the physiology and morphology of the host. However, it is important to note that some rust resistance genes may exhibit reduced efficacy at elevated temperatures associated with climate change. However, the majority of disease prediction models that have been studied so far are subject to the effect of several biotic and abiotic factors that affect the host, pathogen, and their interactions, as well as the specific types of disease prediction models used [187]. Therefore, it might be argued that making definitive conclusions based merely on a limited number of simulation studies would be insufficient for breeders seeking to develop resistance to a particular plant disease. The modification of genetic resistance in host plants to counteract pathogen infections is a prominent issue in the context of climate change and its impact on various host–pathogen interactions. Alterations in the host morphology and physiology due to climate change have a direct association with disease resistance expression. This relationship may be leveraged for the improvement of disease resistance in wheat via the use of both conventional breeding techniques and genetic-engineering breeding tools [188].

## 7. Future Prospects and Limitations

Clearly, the effective management of leaf rust necessitates a basic understanding of the diversity and virulence profiles of the pathogen populations acquired using a race survey analysis approach. These surveys aid in predicting the occurrence of epidemics and provide valuable insights that may be effectively used by breeders and agronomists for integrated disease management strategies. Accumulating evidence has reported quantitative variations among rust fungi isolates belonging to the same race phenotype. The elucidation of the underlying genetics of traits such as the efficiency of infection, latency period, and sporulation rate will provide valuable insights into the mechanisms that are responsible for their manifestation. The complexity associated with measuring these phenotypes is one of the greatest hurdles. 

The identification of some genes linked to either prehaustorial resistance or posthaustorial resistance can be restricted in gene ontology terms; therefore, it was suggested that, to identify appropriate markers and discover genes within the background of prehaustorial resistance, it is necessary to have a segregating population [100]. The prospective utilization of molecular-assisted selection (MAS), specifically through the employment of the SNP marker SNP_1364455, presents an opportunity to transfer the resistance of PI272560 to existing high-quality wheat varieties. This approach holds promise for the establishment of a non-host resistance characterized by prolonged durability and independence from specific pathogen races [100]. The use of einkorn wheat as a genetic resource in wheat breeding has been documented [189]. However, the scarcity of comprehensive genetic and genomic datasets pertaining to einkorn wheat, as well as its genome organization, is a significant challenge in the field of wheat breeding, given the paramount significance of einkorn wheat in this domain. Recently an einkorn genome database was unveiled, and it ushers in an interface for the research community that focuses on cereals to enhance their breeding programs by using comparative genomic and applied genetics [190]. While *T. monococcum* is not the immediate progenitor of the A-genome in bread wheat, it exhibits significant homology to the A-genome seen in present cultivated hexaploid and tetraploid wheat, and gene transfers between bread wheat and *T. monococcum* are indeed possible. Therefore, the utilization of diploid wheat accessions conferring high prehaustorial resistance to leaf rust may broaden the durable resistance pool against leaf rust, and may also be used to replace the commonly used race-specific and single-gene resistance.

Further investigation is required to explore the roles of important SAR regulators, such as NPR3, NPR4, and WRKY transcription factors, in wheat. The downstream genes implicated in these biological processes, particularly *PR* genes, have shown significant promise as transgenic assets for enhancing the broad-spectrum resistance of wheat against many pathogens, indicating that additional research efforts may be required to enhance the cloning and characterization of newly discovered *PR* genes. Investigating the intricate interplay of gene pathways associated with various SAR-like responses is a challenging but significant endeavor. Understanding the sources and distribution patterns of leaf rust resistance genes has significant importance in the development of novel wheat cultivars with durable resistance. The wild relatives of wheat continue to be very significant sources for the identification of novel genetic loci that contain the *Lr*/*QLr* genes. The identification of novel quantitative trait loci associated with various *Pt* races can aid future wheat-breeding programs through the recombination of different loci for durable resistance to leaf rust races. Therefore, there is still a need to explore resistant germplasm, especially introgression lines derived from wheat wild relatives. Further investigation is required to ascertain the optimal approaches for incorporating the rapidly advancing knowledge from several disciplines into effective regional breeding initiatives. The wild relatives of wheat provide a greater reservoir of *R* genes because they have not undergone the genetic bottleneck feature of domestication.

## 8. Conclusions

In this review, we discussed some of the wheat resistance mechanisms to leaf rust, which is one of the most devastating diseases that continuously threatens wheat production. An understanding of plant defense mechanisms against pathogens is fundamental in protecting the global food supply, as well as developing durable, highly disease-resistant plant species. This review will facilitate a more holistic understanding of the complexity of wheat leaf rust.

## Figures and Tables

**Figure 1 plants-12-03996-f001:**
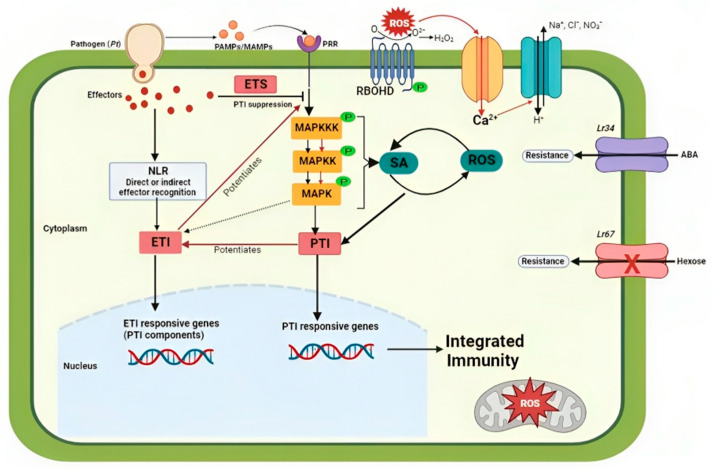
Schematic representation of wheat response to *Pt* infection. Recognition of pathogen-derived conserved molecules (PAMPs/MAMPs) by pattern recognition receptors (PRRs) activates PAMP-triggered immunity (PTI). The earliest events of PAMP recognition include influx of Ca^2+^ ions followed by opening of the membrane transporters for the influx of H^+^ and efflux of K^+^, Cl^−^, and NO_3_^−^ that results in extracellular pH changes and plasma membrane depolarization. Subsequent phosphorylation of downstream components such as respiratory burst oxidase homolog D (RBOHD) and MAPKKK triggers a reactive oxygen species (ROS) burst, Ca^2+^ influx, MAPK activation, phytohormone (salicylic acid—SA) production. Ca^2+^-dependent or Ca^2+^-independent phosphorylated RBOHD produces ROS that lead to a further increase in cytosolic Ca^2+^ concentrations. Pathogens induce susceptibility by interfering with the immune signaling network through the secretion of effectors, resulting in effector-triggered susceptibility (ETS). Direct or indirect recognition of effectors by plant wheat R proteins activates host defense responses, which suppress invasive pathogen growth and proliferation, and this is called effector-triggered immunity (ETI). ETI resistance and responses are dependent on PTI pathway components, and ETI potentiates and restores PTI through upregulation of PTI components. This implies that the two signaling cascades function in a cohort to ensure effective immunity. Lr34, an adenosine triphosphate-binding cassette (ABC) transporter, and Lr67, a sugar transporter (STP) protein, confer multi-pathogen resistance through regulation of abscisic acid (ABA) and hexose sugar molecules, respectively.

**Figure 2 plants-12-03996-f002:**
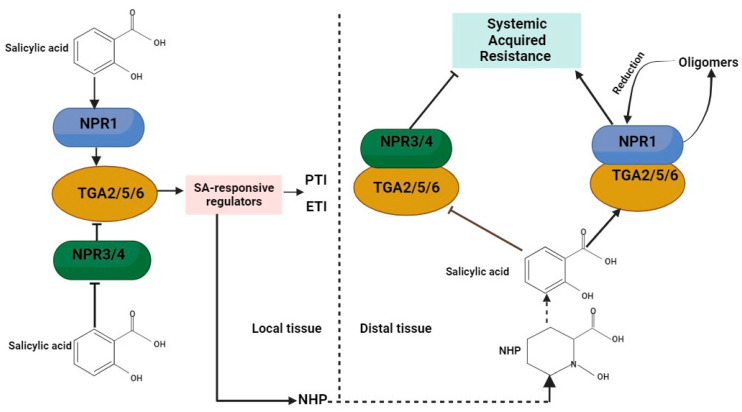
Role of salicylic acid (SA) receptors in plant immunity. SA is perceived by two classes of receptors: NPR1 and NPR3/NPR4. NPR3 and NPR4 interact with TGA2/TGA5/TGA6 in the absence of SA to repress downstream defense gene expression. Binding of SA abolishes the transcriptional repression activity of NPR3/NPR4 and enhances the transcriptional activation activity of NPR1, leading to the upregulation of SA-responsive defense regulators. In local tissues, the expression of SA-responsive defense regulators promotes both PTI and ETI and stimulates the production of the systemic acquired resistance mobile signal N-hydroxypipecolic acid (NHP) by activating the expression of NHP biosynthetic genes. In distal tissues, NHP promotes SA biosynthesis and SA-induced defense response.

**Table 1 plants-12-03996-t001:** *Pt* effectors involved in the manipulation of host reactions.

Effector Protein	Host	Localization	Function in Virulence	References
Pt2567	Wheat	Secretion pathway	Inhibits programmed cell death and serves a non-toxic role in the infection of *TcLr28*.	[58]
Pt3	Wheat	Unknown	Function in avirulence against wheat leaf rust in resistant genotypes.	[59]
Pt27	Wheat	Unknown	Functions in avirulence against wheat leaf rust in resistant genotypes.	[59]
Pt18906	Wheat	Nucleus and cytoplasm	Acts in the cytoplasm and may cause accumulation of reactive oxygen species and callose in TcLr10+27+31.	[55]
Pt13024	Wheat	Nucleus and cytoplasm	Inhibits programmed cell death and triggers reactive oxygen species accumulation and callose deposition.	[56]
Pt_21	Wheat	Apoplast	Suppresses host defense responses by directly targeting wheat TaTLP1 and inhibiting its anti-fungal activity.	[57]

**Table 2 plants-12-03996-t002:** A summary of cloned *Lr* genes for leaf rust resistance.

Gene	Chromosome Position	*R*-Gene Product	*R*-Gene Class	Cloning Technique	References
*Lr1*	5DL	NLR	ASR	Map-based cloning	[104]
*Lr9/Lr58*	6BL/2BL	Tandem kinase–von Willebrand factor type-A domain fusion	ASR	MutIsoSeq	[114]
*Lr10*	1AS	NLR	ASR	Map-based cloning	[105]
*Lr13/Ne2*	2BS	NLR	APR	MutRenSeq	[110,111]
*Lr14a*	7BL	Ankyrin transmembrane domain protein	ASR	MutChromSeq	[113]
*Lr21*	1DL	NLR	ASR	Map-based cloning	[106]
*Lr22a*	2DS	NLR	APR	Map-based cloning and TACCA	[112]
*Lr34/Yr18/Sr57*	7DS	ATP-binding cassette transporter	APR	Map-based cloning	[107]
*Lr42*	IDS	NLR	ASR	BSR-Seq mapping	[108]
*Lr47*	7AS	NLR	ASR	Map-based cloning EMTA approaches	[115]
*Lr67/Yr46/Sr55*	4DL	Anion transporter	APR	Map-based cloning	[109]

NLR—Nucleotide-binding site leucine-rich repeat, ASR—all-stage resistance, APR—adult plant resistance.

## Data Availability

Not applicable.

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
