# Peer review of "The Underexplored Mechanisms of Wheat Resistance to Leaf Rust"

_plants, 2023, doi:10.3390/plants12233996_

Round 1
Reviewer 1 Report
Comments and Suggestions for Authors
The review is well written with suitable citations of references, and illustrations. The authors can present a table describing the genes and their function for different types of resistance to leaf rust in wheat.
Author Response
Response to Reviewer 1 Comments
Point 1: The authors can present a table describing the genes and their function for different types of resistance to leaf rust in wheat.
Response 1: Thank you for your suggestion
We have presented the table that shows the wheat leaf rust resistance genes that have been cloned and functionally characterized so far as indicated from Line 275 – Line 296
Reviewer 2 Report
Comments and Suggestions for Authors
Over all comment about this article
Comments and Suggestions for Authors
It is pleasure to review the manuscript entitled “The underexplored mechanisms of wheat resistance to leaf rust” submitted to plants. Authors of this article reviewed several articles related to wheat diseases and identified the challenges from stresses and discussed from the understanding of latest advancement in the development of wheat resistance to leaf rust. Authors claimed that the available disease control approaches are mostly inactive that lead to the failure of vertical resistance. They also claimed that, little research has been done on prehaustorial and posthaustorial resistance to wheat leaf rust. Therefore, with summarization of some underexplored mechanisms for wheat resistance, they suggested that, comprehensive understanding of wheat defense mechanisms is needed for the development and maintenance of cultivars with resistance to diseases.
Overall, the reviewed results are important and convincing. Thus, the presented information takes up an important topic consistent with the profile of the Journal. However, it needs some improvements. After this improvement, the article will have a better version for the publication. I hope my suggestions can improve the manuscript and will make also important to the wider readers.
L21-25, and L82-87—almost redundant.
3. Introduction: Nicely presented.
However, authors should consider the most update information as well as references in this section. Introduction does not imply completely the title. Need some discussion about what are the underexplored mechanisms? Also need discussion about prehaustorial and posthaustorial resistance, systemic acquired resistance with relevant references. What are limitations of previous works? Why this review is necessary? There are huge works on leaf rust and pt and many varieties have been developed for several crops with pt resistance, therefore, pt related discussion is necessary in this review.
2.2. Wheat innate immunity confers resistance to Pt: I gone through the title of the references described in this section [33-51]. From the title of the references [33-51] I could not see anyone imply wheat, leaf rust, pt; those might be for may other crops, does it directly relate to wheat and leaf rust? If not, you need to rewrite this section. Section subtitle and discussion not relevant. You may change subtitle.
L140: Fig. 1 [48]—But ref 48 did not discuss this. It is in Arabidopsis
L184: As you mentioned “very few papers have been published about prehaustorial and posthaustorial resistance” this indicating that these topic could not make attention to researchers. Therefore, you need to discuss first, why research on prehaustorial and posthaustorial resistance is necessary. Also discuss limitations and difficulties of previous works. Give clues for new windows of research.
4. Systemic acquired resistance---------------it should be 3.3. under section 3. Mechanisms of wheat resistance to leaf rust
Moreover, no recommendation of the further use of those strategies. Not much novel suggestions. No hypothesis is suggested.
-Please discuss, how this field could be impacted in the future research that may be of significance to the scientific community.
-Future prospects and limitations should be addressed.

Comments on the Quality of English LanguageExtensive editing of English language required
Author Response
Response to Reviewer 2 Comments
Point 1: L21-25, and L82-87—almost redundant.
Response 1: Thank you for your suggestion
We have corrected
Point 2: Authors should consider the most update information as well as references in this section. Introduction does not imply completely the title. Need some discussion about what are the underexplored mechanisms? Also need discussion about prehaustorial and posthaustorial resistance, systemic acquired resistance with relevant references. What are limitations of previous works? Why this review is necessary? There are huge works on leaf rust and pt and many varieties have been developed for several crops with pt resistance, therefore, pt related discussion is necessary in this review.
Response 2: Thank you for your suggestion
The introduction has been rewritten and all the issues raised by the reviewer has been addressed. Also, some sections discussion Pt host range and profile as well as the issue of coevolution has been discussed in Sections 2.1 and 2.2 that were added from Line 100– Line 152 .
Point 3: Wheat innate immunity confers resistance to Pt: I gone through the title of the references described in this section [33-51]. From the title of the references [33-51] I could not see anyone imply wheat, leaf rust, pt; those might be for may other crops, does it directly relate to wheat and leaf rust? If not, you need to rewrite this section. Section subtitle and discussion not relevant. You may change subtitle.
Response 3: Thank you for your suggestion
We have corrected it and the subtitle has been changed, the section has been rewritten as per suggestion including the relevant information and references as indicated from Line 181 – Line 252
Point 4: L140: Fig. 1 [48]—But ref 48 did not discuss this. It is in Arabidopsis
Response 4: Thank you for your suggestion
We have removed the reference; it wasn’t supposed to be there.
Point 5: L184: As you mentioned “very few papers have been published about prehaustorial and posthaustorial resistance” this indicating that these topic could not make attention to researchers. Therefore, you need to discuss first, why research on prehaustorial and posthaustorial resistance is necessary. Also discuss limitations and difficulties of previous works. Give clues for new windows of research.
Response 5: Thank you for your suggestion
We have addressed the concerns as indicated from Line 254 – Line 274
Point 6: Systemic acquired resistance should be 3.3. under section 3. Mechanisms of wheat resistance to leaf rust
Moreover, no recommendation of the further use of those strategies. Not much novel suggestions. No hypothesis is suggested.
-Please discuss, how this field could be impacted in the future research that may be of significance to the scientific community.
-Future prospects and limitations should be addressed.
Response 5: Thank you for your suggestion
We have moved the systemic acquired resistance to the section under mechanisms of resistance. Some recommendations and suggestions were made under the section of future prospects and limitations as indicated from Line 590 – Line 640

Reviewer 3 Report
Comments and Suggestions for Authors
This review paper described the mechanisms underlying wheat resistance to Pt infection including prehaustorial and posthaustorial resistance, systemic acquired resistance, and the role of reactive oxygen species with graphical and evidently. I'm happy to accept this review paper.
Author Response
Response to Reviewer 3 Comments
Point 1: This review paper described the mechanisms underlying wheat resistance to Pt infection including prehaustorial and posthaustorial resistance, systemic acquired resistance, and the role of reactive oxygen species with graphical and evidently. I'm happy to accept this review paper.
Special thanks to you for your good comments.

Reviewer 4 Report
Comments and Suggestions for Authors
Dear Authors,
The manuscript is well written, but an automated online, well-known plagiarism check tool in the abstract detected plagiarism; perhaps it is false positive detection, thus I can't comment further about the automated detection of plagiarism. Please also check yourselves. However, I trust the authors; please clear this issue, and I recommend having a substantial revision of the manuscript.
Further, I made the following comments to improve your manuscript:
When the authors saying about population of rust pathogen, whether they intend to say about more virulent fungal pathogen strains of this species?
When the authors are discussing random mutations, it is also better to focus on specific aspects concerning the emergence of virulent genes from newly emerging fungal strains due to co-evolution. I suggest authors add some relevant aspects of co-evolution.
DAMP is Damage-Associated Molecular Patterns (DAMPs), isn't it? not Damaged-Associated Molecular Patterns (DAMPs), please check for typos and grammatical errors.
I think the plant's immune system can't be activated spontaneously, and the constitutive expression of genes associated with the immune system is limited, despite the fact that it lacks adaptive immunity. Please make it clear about your presentation, and please quote an appropriate reference. Please read these articles:
https://nph.onlinelibrary.wiley.com/doi/10.1111/nph.15596
https://doi.org/10.1016/j.pbi.2021.102045
Like these genes, Lr34 and Lr67 confer race-specific resistance against multiple pathogens and adult plant resistance. Whether prehaustorial resistance is specifically based on the first layer of the basal immune system through callose deposition, etc. and why do these kinds of genes cause adult plant resistance rather than conferring resistance at the juvenile stage? Is there a relationship between the facile invasion of haustorium in the early stages of plants and adult plants?

Comments on the Quality of English LanguageMinor English language edits are required.
Author Response
Response to Reviewer 4 Comments
Point 1: The manuscript is well written, but an automated online, well-known plagiarism check tool in the abstract detected plagiarism; perhaps it is false positive detection, thus I can't comment further about the automated detection of plagiarism. Please also check yourselves. However, I trust the authors; please clear this issue
Response 1: Thank you for your suggestion
We have rechecked it and we think it’s false positive detection because the abstract was completely synthesized by the authors.
Point 2: When the authors saying about population of rust pathogen, whether they intend to say about more virulent fungal pathogen strains of this species?
Response 2: Thank you for your suggestion
We are referring to population diversity – Line 39
Point 3: When the authors are discussing random mutations, it is also better to focus on specific aspects concerning the emergence of virulent genes from newly emerging fungal strains due to co-evolution. I suggest authors add some relevant aspects of co-evolution.
Response 3: Thank you for your suggestion
We have added a section discussing the aspects of coevolution – Line 100 – 152
Point 4: DAMP is Damage-Associated Molecular Patterns (DAMPs), isn't it? not Damaged-Associated Molecular Patterns (DAMPs), please check for typos and grammatical errors.
Response 4: Thank you for your suggestion
We have removed it
Point 5: I think the plant's immune system can't be activated spontaneously, and the constitutive expression of genes associated with the immune system is limited, despite the fact that it lacks adaptive immunity. Please make it clear about your presentation, and please quote an appropriate reference.
Response 5: Thank you for your suggestion
We have corrected and appropriate references have been added as well – Line 46 – Line 55
Point 6: Like these genes, Lr34 and Lr67 confer race-specific resistance against multiple pathogens and adult plant resistance. Whether prehaustorial resistance is specifically based on the first layer of the basal immune system through callose deposition, etc. and why do these kinds of genes cause adult plant resistance rather than conferring resistance at the juvenile stage? Is there a relationship between the facile invasion of haustorium in the early stages of plants and adult plants?
Response 6: Thank you for your suggestion
Lr34 and Lr67 confer non-race specific resistance, also referred to as adult plant resistance against multiple pathogens. We have added relevant literature and references about this resistance mechanism as indicated in line 354 – 372. We have added relevant literature and references. Functional targets of prehaustorial resistance mechanisms are germ tube and appressorium (ectophytic phase of pathogenesis), as well as substomatal vesicle and infection hypha (early endophytic phase of pathogenesis of rust fungi). This defense response typically leads to the formation of cell wall reinforcements, also called cell wall appositions or papillae.

Round 2
Reviewer 2 Report
Comments and Suggestions for Authors
The manuscript has been sufficiently improved based on suggestions.
Comments on the Quality of English LanguageMinor editing of English language required
Author Response
Point 1: Minor editing of English language required
Response 1: Thank you for your suggestion
The English language has been edited

Reviewer 4 Report
Comments and Suggestions for Authors
Dear Authors,
I suggest authors rewrite the abstract. Please submit a clearly revised manuscript instead of a track-change manuscript.
Author Response
Point 1: I suggest authors rewrite the abstract. Please submit a clearly revised manuscript instead of a track-change manuscript.
Response 1: Thank you for your suggestion
The abstract has been rewritten and the track-changes have been removed.
